# The Landscape of Nanovectors for Modulation in Cancer Immunotherapy

**DOI:** 10.3390/pharmaceutics14020397

**Published:** 2022-02-11

**Authors:** Simona-Ruxandra Volovat, Corina Lupascu Ursulescu, Liliana Gheorghe Moisii, Constantin Volovat, Diana Boboc, Dragos Scripcariu, Florin Amurariti, Cipriana Stefanescu, Cati Raluca Stolniceanu, Maricel Agop, Cristian Lungulescu, Cristian Constantin Volovat

**Affiliations:** 1Department of Medical Oncology-Radiotherapy, “Grigore T. Popa” University of Medicine and Pharmacy, 16 University Str., 700115 Iaşi, Romania; simonavolovat@gmail.com (S.-R.V.); dianaiboboc@gmail.com (D.B.); floryn_ciprian@yahoo.com (F.A.); 2Department of Radiology, “Grigore T. Popa” University of Medicine and Pharmacy, 16 University Str., 700115 Iaşi, Romania; corina.ursulescu@gmail.com (C.L.U.); lgheorghe123@gmail.com (L.G.M.); cristian.volovat@yahoo.com (C.C.V.); 3Department of Medical Oncology, “Euroclinic” Center of Oncology, 2 Vasile Conta Str., 700106 Iaşi, Romania; 4Department of Surgery, “Grigore T. Popa” University of Medicine and Pharmacy, 16 University Str., 700115 Iaşi, Romania; dscripcariu@gmail.com; 5Department of Biophysics and Medical Physics-Nuclear Medicine, “Grigore T. Popa” University of Medicine and Pharmacy, 16 University Str., 700115 Iaşi, Romania; cipriana.stefanescu@yahoo.com (C.S.); catistolniceanu@yahoo.com (C.R.S.); 6Physics Department, “Gheorghe Asachi” Technical University, Prof. Dr. Docent Dimitrie Mangeron Rd., No. 59A, 700050 Iaşi, Romania; m.agop@yahoo.com; 7Department of Medical Oncology, University of Medicine and Pharmacy, 200349 Craiova, Romania; cristilungulescu@yahoo.com

**Keywords:** immunotherapy, nanomedicine, nanotechnology, nanopharmaceuticals, nanoparticles, bioinspired nanovectors

## Abstract

Immunotherapy represents a promising strategy for the treatment of cancer, which functions via the reprogramming and activation of antitumor immunity. However, adverse events resulting from immunotherapy that are related to the low specificity of tumor cell-targeting represent a limitation of immunotherapy’s efficacy. The potential of nanotechnologies is represented by the possibilities of immunotherapeutical agents being carried by nanoparticles with various material types, shapes, sizes, coated ligands, associated loading methods, hydrophilicities, elasticities, and biocompatibilities. In this review, the principal types of nanovectors (nanopharmaceutics and bioinspired nanoparticles) are summarized along with the shortcomings in nanoparticle delivery and the main factors that modulate efficacy (the EPR effect, protein coronas, and microbiota). The mechanisms by which nanovectors can target cancer cells, the tumor immune microenvironment (TIME), and the peripheral immune system are also presented. A possible mathematical model for the cellular communication mechanisms related to exosomes as nanocarriers is proposed.

## 1. Introduction

A malignant cell can harbor more than 11,000 genomic mutations in addition to new tumor-associated antigens (TAAs), including antigens produced by oncogenic viruses, altered glycoproteins, glycolipids, or oncofetal antigens [1]. These new tumor-associated antigens can be presented on cell surfaces along with major histocompatibility complex (MHC) molecules. The role of the immune system in cancer has been underestimated for many decades because tumor cells suppress the immune response by enhancing negative regulatory pathways involved in immune homeostasis or adopting features that prevent detection by the immune system. Two well-known checkpoints are cytotoxic T-lymphocyte protein 4 (CTLA4) and programmed cell death protein 1 (PD-1). CTLA4 has the role of inhibiting T cells, being in competition with the co-stimulatory molecules CD28 and CD86. PD-1 is a cell-surface receptor that is expressed by T cells, binding to the ligands PD-L1 and PD-L2. These ligands are expressed in a variety of cells, though PD-L2 is mainly expressed on dendritic cells in normal tissues [2,3,4,5], and antibodies that inhibit the interaction between PD-L1 and PD-1 produce clinical responses in a wide range of solid and hematologic malignancies [6] (Figure 1).

Cancer immunoediting is a dynamic process consisting of three main phases: elimination, equilibrium, and escape. The elimination phase represents a modern vision of cancer immunosurveillance, where the molecules and cells with innate and adaptative immunity cooperate to identify the presence of developing tumors and eliminate them. Sometimes, variants of tumor cells may not be completely destroyed but enter an equilibrium phase in which the immune system controls tumor cell growth. The components of the immune system that participate in the elimination phase include cytokines (IFN-α/β, IFN-γ, IL-12, and TNF), dendritic cells, macrophages, cells of innate immunity, such as NK or NKT cells, cells of adaptive immunity (CD4^+^ and CD8^+^ T cells), and immune effector molecules (perforin and TRAIL). The mechanisms for alerting the immune system to the presence of a developing tumor have not been fully characterized. It is supposed that a developing tumor stimulates the production of “danger signals”, which are cytokines, such as type I IFNs, that activate dendritic cells, natural killer cells, and macrophages. The equilibrium phase is the second phase of cancer immunoediting, where the innate immune system cannot eliminate cancer cells, but keeps them in a state of immune-mediated tumor dormancy. The tumor cells and host immune system exist in a dynamic balance, where the immune system does not fully eradicate the heterogeneous tumor. Some of the tumor cells evade immune-mediated recognition and destruction [7]. A dramatic result of immunoediting is represented by tumor escape from immune control. The escape phase can be considered as a failure of the immune system to eliminate or control the cancer cells, enabling the survival of cell variants, which grow in an unrestricted manner. The immune phenotypes of the TME include both adaptive and innate immune cells that have a major influence on immunotherapy, and they are classified into three principal phenotypes: the immune-desert, immune-excluded, and inflamed phenotypes. The immune-desert phenotype (lacks antitumor immune cells) is characterized by immunological tolerance (losing the response to antigen presentation), ignorance (lack of antigens), and a lack of T cell priming [8]. These tumors include pancreatic and prostate cancers, and have poor responses to immune checkpoint inhibitors (ICI) and worse outcomes compared with other phenotypes due to the lack of pre-existing cytotoxic T cells and a poor clonal repertory of T cell receptors. In the immune-excluded phenotype, the immune cells from the tumor periphery or stroma are hampered by extravascular stroma and immature vessels. Moreover, the expression of transforming growth factor-β (TGF-β) and the density of cancer-associated fibroblasts (CAFs) are enhanced [9,10]. Tumors of this phenotype are more sensitive to immune checkpoint inhibitors (ICI) than those with immune-desert phenotypes due to the existence of the CD8^+^ T-effector-cell phenotype in the stroma, which can proliferate and become active. The third phenotype is the inflamed phenotype, where proinflammatory cytokines are expressed by T cells from the parenchyma, representing a failure of the antitumor immune response [11]. Although a large number of T cells with receptors against tumor-associated antigens are present, immune cells suppressed by hypoxia are also abundant. Examples of tumors with this phenotype are non-small-cell lung cancer and melanoma. This phenotype is considered to have the most potential in terms of sensitivity to ICI [12].

Superposed on the three main phenotypes of TME described above, the classification of tumors into two categories of “hot” and “cold” tumors was proposed, referring mainly to T cell infiltration, and the classification of these tumors into four categories was recently suggested, namely, into hot, altered–excluded, altered–immunosuppressed, and cold [13]. This concept for patient stratification is related to the type, location, and density of immune cells within a tumor site, and it can provide more accurate information than the classical TNM system for any type of cancer [14,15]. The classification into “hot” and “cold” tumors led to the development and implementation of the Immunoscore, which is a robust, consensus-based, standardized scoring system [16,17,18]. Cancer immunotherapy is focused on developing agents that promote strategies for the recognition and destruction of tumor cells by the immune system and represents a new alternative to classical therapies [19,20,21,22].

Classical cancer immunotherapy can be classified into (a) synthetic immunotherapy, involving programming to initiate new immune responses directed toward targets expressed by tumors, such as monoclonal antibodies (MoAbs) and chimeric antigen receptors (CARs), and (b) molecules designed to enhance natural immune responses, such as immune checkpoint inhibitors (ICIs) [23].

Compounds targeting PD-1 and CTLA-4 are the best-known immune checkpoint inhibitors that suppress T cell responses to cancers and target the tumors to enable antitumor immunity. To date, 14 different immunomodulators—seven checkpoint inhibitors, four cytokines, two adjuvants, and a small molecule with immunomodulatory properties—have been approved by regulatory agencies (FDA, EMA, NICE) for the treatment of more than a dozen major cancer types. 

Other ICI therapies are currently in various stages of clinical testing for many different tumor types. Camrelizumab, pidilizumab, sintilimab, BMS-936559 (MDX-1105), and toripalimab (JS001) are some examples of those undergoing clinical trials and being investigated for their efficacy and safety profiles [24,25,26,27]. 

## 2. Mechanisms of Resistance to Immune-Checkpoint Blockades in Cancer

The heterogeneity of tumors and the complex immune microenvironment represents an important issue for treatment efficacy and is related to variations in the immune system that occur from individual to individual [28]. Immunotherapeutic resistance is classified as either primary (intrinsic) or acquired (extrinsic) resistance. Primary resistance represents a non-response of cancer to an immunotherapeutic strategy [29,30]. Intrinsic resistance involves the hyperprogressive diseases (HPDs) that causes the alteration of chromosome 11 region 13, (MDM)2/4 gene amplification, and epidermal growth factor receptor (EGFR) gene mutation [31,32]. TME alterations (polarization of macrophages) and a low tumoral mutational burden (TMB) are other factors influencing resistance [33,34,35]. The extrinsic mechanisms of resistance to immunotherapy are related to tumor-infiltrating lymphocytes (TILs) in the TME [36,37,38]. It was reported that the infiltration of immunosuppressive cells (Tregs, MDSCs, TAMs) in the TME is always associated with immunosuppression [39]. The “cold tumors” characterized by immunosuppressive tumor stroma are usually associated with a low mutational burden and a low neoantigen presence [40]. Tumor-associated TLSs are associated with good prognosis in the majority of cancer types, demonstrating the possibility to promote a systemic and long-lasting antitumor response [41].

## 3. Bioactive Nanoparticles Designed to Modulate Cancer Immunotherapy

Nanoscience is considered an “enabling technology” that impacts various fields of research and everyday life. This term initially included nanomedicine, also termed nanotechnology, which includes nanoparticles (NPs), but the expansion of the field of molecules that carry medicines has led to the emergence of the new and vastly more appropriate term nanovectors. The specific targeting of cancer tumors is the key to increasing treatment efficacy while reducing detrimental off-target effects and represents a major scientific issue. In recent years, vectorization approaches have expanded with the discovery of new families of nanovectors (with dimensions from 1 to 1000 nm) created by chemical engineering (e.g., nanoparticles) or related to the biological world (e.g., viruses, bacteria, and extracellular vesicles) [42].

Drug development has become more complex, and efficient vectorization will improve the safety and efficacy of cancer therapies, representing a turning point in cancer treatment, experiencing a second birth after having been neglected for years. An interplay between cancer nanomedicine and immunotherapy can actually be described and has been demonstrated in multiple preclinical studies. Various types of materials have been tested for biomedical applications, including polymers, lipids, carbon structures, metals, and other organic and inorganic materials that can be used for nanopharmaceutical formulations, all of which have different features. The delivery of bioactive molecules using NPs has the potential to meet the goals of increasing the therapeutic efficacy and reducing the side effects of these molecules through improved pharmacokinetics and biodistribution. 

There are various applications for which NPs can be used to enhance the efficacy of cancer immunotherapy. These include the delivery of antigens and adjuvants as vaccines, and the delivery of molecules and antibodies targeting specific cells, such as APCs or dendritic cells whose interaction modifies the tumor microenvironment.

The sizes of nanoparticles must range from 1 to 100 nm, and they must also have high surface-area-to-volume ratios and advantageous delivery kinetics [43]. Small nanoparticles (<10 nm) can be frequently cleared by the kidneys, whereas larger nanoparticles, larger than 200 nm, are more likely to be fenestrated in the spleen in addition to showing variable intratumoral distribution depending on regional blood flow. In addition, nanoparticles must have non-antigenic coatings to avoid triggering an immunogenic response in the host, as well as an enhanced ability to accumulate via passive targeting into highly angiogenic tumors. Another critical design parameter is the particle shape, which influences how a nanoparticle moves within the blood circulation, enters the cells, and stimulates an immune response [44]. Initial formulations were composed of nanoparticles with spherical shapes, but later advantages in nanoparticle engineering rendered the emergence of a new portfolio of possible shapes that include rods, prisms, cubes, stars, and disks. Asymmetric nanoparticles can also be manufactured and may show such advantages as enhanced nanoparticle penetration and distribution inside solid tissues and tumors [43]. It has been suggested that the Th1/Th2 polarization of the immune response is influenced by the particle shape. An important issue is related to the charging of NPs by loading moieties onto the particle surfaces via electrostatic interactions. Cationic nanoparticles can generate acute systemic toxicity and stimulate acute inflammation. A suggested mechanism could be that the cationically charged polymers used to construct nanoparticles can trigger pattern-recognition receptors in immune cells [45]. Another property of NPs that plays a critical role in their biodistribution, cellular uptake, cellular association, and immune response is ligand density. Ligand conjugation to a nanoparticle, like the functionalization of mesoporous silicon nanoparticles with amines, can reduce systemic cytotoxicity [46]. In folate-targeted liposomes, it was observed that the internalization and externalization rates for a targeted receptor affected the optimal ligand density, which is critical for maximizing nanoparticle uptake. The flexibility of a nanoparticle can also modify antibody-mediated targeting, phagocytosis, and endocytosis. It was reported that particle endocytosis occurred more rapidly with flexible nanoparticles [47,48,49]. In conclusion, certain characteristics must be taken into account when designing an appropriate nanoparticle. 

### 3.1. Shortcomings in Nanoparticle Delivery and Efficacy 

The main factors that influence the targeting of NPs in tumors are the physiochemical properties of nanoparticles, which are influenced by such factors as the size; the shape; coating with tumor cell-targeting antibodies, aptamers, peptides, and/or small molecules that are able to interact with malignant cells; the properties of the tumor, such as the tumor type, size, and stage; and the influence of the mononuclear phagocytic system (MPS). The MPS comprises the spleen, liver, bone marrow, lymph nodes, skin, and other organs that contain resident phagocytic cells, such as macrophages. Macrophages from the MPS organs are derived from circulating monocytes. The nanoparticles are sequestered mainly by the liver and the spleen.

In the liver, the macrophages include Kupffer cells and motile macrophages and are located in liver sinusoids. The NPs flowing through the liver capillaries are recognized by the scavenger receptors of Kupffer cells and engulfed by them. The types and chemistry of NPs decide their fate; large inorganic NPs reside in Kupffer cells for a long time, whereas organic particles are rapidly degraded and eliminated [50]. The spleen contains macrophages that are involved in erythrocyte degradation in the red pulp, while the white pulp contains metallophilic macrophages and is involved in clearing apoptotic cells [51]. The main mechanism by which macrophages sequester NPs is phagocytosis, but this can also occur by clathrin- or caveolin-mediated endocytosis [52]. Macrophages sequester nanoparticles by phagocytosis, micropinocytosis, endocytosis, and other mechanisms. Low-density lipoproteins, nanoparticles, and bacteria are taken up through scavenger receptors [53].

Delivery issues are a major barrier to nanoparticle carriers, which encounter various physical and biological barriers (e.g., flow and shear forces, diffusion, phagocytic activity, and renal clearance) that influence their access to target tissues and cells [54,55,56,57,58].

The mononuclear phagocytic system (MPS) and the renal clearance pathway represent two factors that influence the accumulation of nanocarriers in tumor cells. The MPS can be defined as a network of organs (mainly the liver and spleen) containing phagocytic cells that take up nanoparticles, while the renal system is a filter that blocks nanoparticles larger than 5.5 nm in diameter. 

Wilhelm et al. reported that a median of 0.7% of the injected dose (ID) of nanoparticles accumulates into a tumor, suggesting that only 7 out of 1000 injected nanoparticles effectively target a solid tumor in a mouse model [59].

The delivery of nanoparticles to tumors is influenced by specific (active) and nonspecific (passive) targeting. Specific targeting involves the functionalization of the nanoparticle surface using ligands that act as alternative target sites, and targets may include tumoral blood vessels, the extracellular matrix, or intracellular targets.

Nonspecific targeting is based on the coating of the nanoparticle with anti-fouling and/or stabilizing agents. The actual view is that nanoparticles cross the tumor vascular barrier through intercellular gaps and become trapped in the tumor due to pressure generated by poor lymphatic drainage—a process termed “enhanced permeability and retention” (EPR)—and are further intratumorally retained via active targeting [60,61].

Once nanoparticles arrive in tumor blood vessels, some of them extravasate into the tumor microenvironment.

When nanoparticles are administered to tumor-bearing animals, they rapidly pass from the systemic circulation into the tumor vasculature, which is highly abnormal, with zones of both rich and poor vascular density, hierarchical disorganization, irregular branching and a serpentine structure, and vascular malformations with arteriovenous shunts [62]. The tumor vascular density is generally the highest at the tumor/host interface; in contrast, the central portions of tumors tend to be less well vascularized and often exhibit zones of necrosis owing to insufficient blood supply [62]. At least five distinctly different types of tumor blood vessels have been described: mother vessels (the first angiogenic type of blood vessel with abnormal hyperpermeability to plasma proteins), feeding arteries, glomeruloid microvascular proliferations (GMPs), capillaries, and draining veins.

Nanoparticles cross mother vessels and enter the tumor compartment, where the blood flow is sluggish, allowing the diffusion of nanoparticles out of the vessel and into the extracellular matrix of the tumor [63]. Some mechanisms for the extravasation of nanoparticles inside the tumor have been described. One is intercellular extravasation, where nanoparticles extravasate from the tumor blood vessels into the tumor microenvironment through gaps between endothelial cells. There is an alternative hypothesis (transcellular extravasation) where nanoparticles can extravasate into tumors via a transendothelial cell pathway [64]. The possibility of studying the transport of nanoparticles through intercellular gaps via the EPR effect has been heavily emphasized.

Once the nanoparticles have crossed the vascular barrier, they penetrate the tumor microenvironment, where they have to overcome biological barriers. After extravasation from the tumor vasculature, nanoparticles interact with components of the tumor stroma, such as nonmalignant fibroblasts, immune cells, and pericytes, and with elements of parenchyma (tumor cells). The heterogeneity of tumors is related to the parenchyma and stromal cells, and also to the ratio of support and secreted proteins. The pressure of the interstitial fluid in the tumor is 10–40 times higher than that in normal tissues, thus generating pressure gradients and heterogeneous flow in the interstitium. This pressure can modify the distribution and transport of macromolecules, drugs, and nanoparticles inside the tumor [65,66,67]. The increased pressure of the interstitial fluid causes high flow of interstitial fluid to the stroma and lymphatic vessels, and this process is strongly correlated with lymphangiogenesis, invasiveness, and metastasis [68].

### 3.2. Modulation of Nanovector Efficacy

The major factors that modulate the efficacy of nanopharmaceuticals are protein coronas, microbiome modulation, and the EPR effect. Protein coronas (PCs) refer to the inappropriate absorption of proteins onto the surfaces of NPs, which results in the NPs having different biological identities. These different identities are responsible for the failure of nanoparticle-based immunotherapy, and PC formation results in the generation of two types of responses: a nonresponse (immune blinding), promoted by the partial or total coverage of the antigens by the PC, and an uncontrolled response (immune reactivity), with a hyperresponse of the immune system against the NP [69,70]. Immune blinding was described by Shanehsazzadeh et al., who observed that, in an in vivo mouse model, NPs showed higher distribution in the blood and muscle than in tumors. The conclusion was that the targeting molecules are covered by PCs, resulting in reduced in vivo tumor uptake, and the degree of immune blinding is sometimes related to the different PCs that are formed. PEGylation can reduce, but not eliminate, PC formation [71,72]. The immune-blinding process can be avoided by changing the physicochemical properties of the NPs [73]. The uncontrolled (immune reactivity) response involves the triggering of excessive immune activity, which is commonly related to a high production of proinflammatory cytokines or complement (C3) activation [74,75]. Biocompatible materials, such as zwitterionic polymers or hydrophilic nanoparticles, can be used to decrease protein adsorption and thus avoid complement activation [76].

Recently, cell-membrane coatings were developed to enable the camouflage of NPs to avoid immune clearance and allow complement activation [77]. A cell membrane coating based on red blood cell (RBC) membranes or PEG camouflages the particles from macrophage uptake, favoring their circulation for longer periods and thereby increasing their chances of accumulating in the tumor [78,79,80].

After being injected into the body, nanoparticles undergo a transformation of molecular identity from a synthetic identity to a biological identity, which includes an acquisition, after the interaction with the body’s fluids of new physicochemical properties [81]. The new biological identity causes certain interactions with immune system cells, especially macrophages [82].

The heterogeneity of the response to immune treatment can sometimes be explained by the influence of gut microbiota, with the supposition that a large number of microorganisms can have a modulatory effect on the functions of immune cells, especially in the case of Tregs and CD4^+^ and CD8^+^ T cells. Some bacteria, such as *Bifidobacterium longum*, *Bifidobacterium adolescentis*, *Lactobacillus* species, and *Parabacteroides merdae*, have been linked to the response to anti-PD-1 treatment, in which they are involved in various mechanisms, such as elevating IFN-γ secretion, enhancing DC function, and increasing the number of CD8^+^ tumor-infiltrating T cells [83,84].

The enhanced permeability and retention (EPR) effect was initially studied on inflammation [85] and represents a correlation of the anatomical and pathophysiological features of the host with the characteristics of solid tumors. The features of the host consist of vascular architecture, or an inadequate secretion of various mediators (bradikinine, carbon monoxide, and vascular endothelial growth factor).

The explanation for this phenomenon is that, in order for tumor cells to grow rapidly, they must generate new blood vessels via VEGF or other growth factors, since they are dependent on having a blood supply. The newly formed tumor vessels have poorly aligned and defective endothelial cells with wide fenestrations, lacking a smooth muscle layer, and having impaired receptors for angiotensin II and effective lymphatic drainage. As a consequence, the fluid transport dynamics become abnormal, especially for macromolecular drugs. The cells in the tumor stroma contain cells that play a crucial role in improving the efficiency of EPR-mediated tumor accumulation. Macrophages have a strong influence on the retention of nanomedicines, and tumor-associated macrophages (TAMs) can act as a nanoparticle depot and gradually release the payload to neighboring tumor cells [86].

The applicability of nanopharmaceuticals and bioinspired nanoparticles has been driven by the use of the EPR effect, as this phenomenon includes pathophysiological factors and biological processes encountered within the body. The exchange surface and the half-life in circulation are critical points of the efficiency of nanoparticles. The EPR effect has been used in the development of new strategies that have improved the effectiveness and safety of NP [87,88,89,90,91]. The applicability of nanopharmaceuticals and bioinspired nanoparticles has been driven by the use of the EPR effect, as this phenomenon includes pathophysiological factors and biological processes encountered within the body. The exchange surface and the half-life in circulation are critical points of the efficiency of nanoparticles. The EPR effect has been used in the development of new strategies that have improved the effectiveness and safety of NP [92,93].

## 4. Types of Nanovectors for Improving Cancer Immunotherapy

Nanopharmaceuticals are generally classified according to their physical and chemical properties, such as material type, shape, size, charge, and surface properties, and it is now generally accepted that all of these features influence their kinetics, biodistribution, cellular uptake, immunogenicity, and loading properties.

The nanovectors used in cancer immunotherapy can be separated into nanopharmaceuticals and bioinspired nanovectors. Pharmaceutical nanotechnology consists of nanosized products that can be transformed in different ways to enhance their characteristics, leading to improvements in terms of prolonged circulation, drug localization, drug efficacy, etc. [94].

Nanopharmaceuticals include polymeric NPs, lipid nanocarriers, metal NPs, mesoporous silica NPs (MSNs), exosomes, and carbon nanotubes (CNTs) (Table 1) [95,96,97,98,99]. Bioinspired nanovectors (nanobioparticles) include the following: extracellular vesicles, bacterial minicells, virus-like particles (VLPs), oncolytic viruses, and phage-display nanobioparticles (Table 2) [100,101,102,103].

### 4.1. Nanopharmaceuticals

**PLGA** (poly(lactic-co-glycolic acid)) (Figure 2) is an FDA-approved polymer that is biocompatible and biodegradable, and can be used to encapsulate many biologically active compounds with low toxicity. PLGA microspheres can target the pathways for MHC class I and class II molecules and enhance DC maturation [67]. PLGA nanoparticles are nonspecifically taken up. PLGA nanoparticles have been designed for the transportation of cytokine agonists, siRNAs, or CpG-coated tumor antigens to enhance antigen uptake by DCs and trigger both CTL (CD8^+^) and Th (CD4^+^) immune responses [138,139,140].

Colzani et al. reported the development of an efficient antibody delivery system that includes trastuzumab and doxorubicin in poly(lactic-co-glycolic) acid nanoparticles capable of affecting the regulatory signaling pathways of cancer cells and stimulating ADCC [141]. In vitro results showed that the PLGA nanoparticles were more suitable for targeting DCs than the PLGA microparticles, with a 10- to 100-fold increased efficiency in hD1 release from nanoparticles [142]. NPs were designed with a maximum density of monoclonal antibodies on the surface, which is responsible for the higher interleukin-10 (IL-10) production and enhanced antitumor response. Therefore, PLGA NPs containing antigenic peptides can target DCs for vaccine delivery, followed by the triggering of the cytotoxic T cell immune response, which blocks the immune-escape mechanism of tumor cells [143,144]. Tumor cells develop genetic and epigenetic alterations to prevent recognition and elimination by immune cells, promoting immune evasion. Gold nanoshells and anti-PD-1 peptide (APP)-loaded PLGA nanoparticles were intratumorally administered in one study, and an antitumoral effect at the primary tumor site was demonstrated, achieved in combination with photothermal therapy [145].

**Dendrimers** are broadly branched macromolecules composed of a core and cavities to entrap drugs. Dendrimers have well-defined chemical structures, water solubilities, and polyvalencies, properties that are suitable for drug delivery [146]. The direct interaction of dendrimers and immune cells has been described. Poly(phosphorhydrazone)dendrimers were found to selectively elevate the proliferation of natural killer cells with anticancer activity [64]. Dufes et al. reported tumor reduction by chemoimmunotherapy based on the use of dendrimers as carriers. The systemic administration of dendrimer nanoparticles containing a TNFA expression plasmid regulated by telomerase gene promoters (hTR and hTERT) results in transgene expression and the regression of remote xenograft murine tumors. In addition, a complex structure designed to contain a CpG oligonucleotide as an immune-stimulating agent and doxorubicin was demonstrated to be targeted by a prostate-specific membrane RNA aptamer [147].

**Lipid nanocarriers** are vesicles containing one or more bilayers of phospholipids that are characterized by high biocompatibility, and the following subcategories are included: solid-lipid nanoparticles, nanoemulsions, lipid nanocapsules, and liposomes. The liposome structure has many similarities with the cell membrane, in which hydrophobic tails of phospholipids cluster together while the heads are hydrophilic. The existence of hydrophobic and hydrophilic compartments allows the encapsulation and delivery of various compounds without affecting their properties, and they are thus considered an ideal drug-delivery system [148,149].

An effective treatment option involving the co-delivery of ovalbumin (OVA) and IFN-encoding pDNA to DCs via liposomes was described. There is a combined therapeutic effect of OVA and IFN-encoding pDNA that enhances the antitumor effect through CTL activation [150]. Using pH-sensitive dextran liposomes, enhanced infiltration of CD8^+^ in the tumor was demonstrated. Curdlan and mannan are bioactive polysaccharides that can be used in the formulation of pH-sensitive liposomes to improve DC activation. The liposomal delivery of cGAMP facilitates the improved activity of STING agonists, resulting in improvements in the immunological memory and rechallenging of tumor cells. Gene delivery for enhancing immunotherapy is represented by the delivery of RNA lipoplex (RNA-LPX) to DC cells that trigger in situ DC maturation. PEGylation is frequently used for the delivery of siRNA. An example is the pH-sensitive cationic lipid YSK05 that was developed as a PEGylated multifunctional envelope-type nanodevice (MEND), and it was demonstrated that this platform enhances gene silencing when administrated intratumorally [151].

**Micelles** are vesicular particles formed by the spontaneous aggregation of amphiphilic molecules, with many applications in cancer treatment as carriers. The synthesis of micelles is easier than that of other nanocarriers, and micelles are also biodegradable, nontoxic, and able to be intracytoplasmically delivered. They are used to carry ovalbumin (OVA) or regulate metabolism-related enzymes, such as IR780, which results in the inhibition of IDO (indoleamine 2,3-dioxygenase), followed by the activation of T lymphocytes and, consequently, the inhibition of distal tumor growth (abscopal effect). Zinc–protoporphyrin IX-grafted polypeptide micelles that target TAMs and stimulate the immune system were designed. The stimulation of T lymphocytes by the repolarization of TAMs was followed by tumor regression [152].

**Gold nanoparticles (GNPs, AuNPs)** can deliver antigenic proteins and gene oligonucleotides to specific sites. The surfaces of AuNPs can undergo covalent and noncovalent interactions with various biomolecules, such as DNA, peptides, and antibodies [153,154,155].

AuNPs were described to influence the nucleus, its subcompartments, and the mitochondria of cancer cells. 

One of the major properties of GNPs is localized surface plasmon resonance (LSPR) that is sensitive to size, material geometry, dimensions, and the dielectric properties of the surrounding media. Consequently, GNPs have been produced in different shapes, such as nanospheres, nanoshells, nanorods (NRs), nanostars, nanocages, and core–shell structures. A series of Au–on–AuNR hybrid structures that are homometallic nanostructures in two new dimensions was also developed; their LSPR was effectively NIR-tuned within the visible NIR (near-infrared region) spectral range, rendering them excellent candidates for photothermal therapy [156].

In photothermal therapy, the absorbed light turns into heat, causing the irreversible distortion of DNA or cells [53]. If nanogold particles are used, there is an increase in light absorption at a certain wavelength, thus reducing the power of the laser for the photothermal removal of cancer cells. GNP-conjugated antibodies can be conjugated to monoclonal antibodies for targeting cancer cells when illuminated by light with a wavelength corresponding to the GNP wavelength [157]. The combination of AuNPs with photothermal ablation is a promising concept that is being researched in numerous different trials. Gold NPs have been used in delivering CgP oligonucleotides to macrophages and DCs followed by a regression in tumor growth. The delivery of adjuvants, such as OVA or CpG, for immunotherapy was carried out using gold nanoparticles of different sizes and shapes, and 15 nm was found to be the size with the best efficacy for the immunotherapeutic delivery of antigens [155].

**Iron oxide nanoparticles** are potent carriers for vaccine delivery. They have a direct effect by polarizing immune cells, such as DCs and macrophages, increasing the immune response, or they can be used as a delivery system with OVA functioning as an immune potentiator [158]. The FDA have approved supplementation with ferumoxytol in mammary cancer due to an intrinsic therapeutic effect. In vitro, it was demonstrated that adenocarcinoma cells incubated together with ferumoxytol and macrophages could enhance caspase-3 activity. Moreover, macrophages exposed to ferumoxytol could induce proinflammatory Th1-type responses in macrophages [159].

**Mesoporous silica NPs (MSNs)** are solid materials with a honeycomb-like porous structure containing hundreds of empty mesopores capable of absorbing large quantities of bioactive molecules [160]. Mesoporous silica materials can show diverse interactions with biosystems, with effects on various properties that contribute to biodegradation, biodistribution, toxicity, cellular uptake, and, more importantly, their interaction with immune cells [160].

Mesoporous silica materials are nontoxic and biodegradable under physiological conditions and can be released to tissues and excreted via renal clearance. Smaller particles and lower concentrations of mesoporous silica were demonstrated to affect smaller proportions of human monocyte-derived dendritic cells (MDDCs) in comparison to the use of larger particles and higher concentrations, which has led to their suggested use as a component of cancer vaccines [161].

A complete vaccine formulation using mesoporous silica (XLMSNs + OVA + CpG-ODN) was developed and successfully induced dendritic cell (DC) maturation with high levels of CD86 expression and increased secretion of proinflammatory cytokines, including IL-12 and TNF-α. MSNs were found to be useful for transporting drugs and siRNAs, which can be co-delivered into the body and induce cytokine secretion [162,163].

**Carbon nanotubes (CNTs)** are cylindrical models composed of carbon that have demonstrated their potential in multiple ways, including as tumor-antigen nanocarriers. Frequently, the CNTs used are multiwalled carbon nanotubes (MWNTs), which were successfully used to co-deliver OVA and CpG to antigen-presenting cells (APCs) [164]. In addition, the photothermal ablation of primary tumors has been achieved with single-walled carbon nanotubes. It was demonstrated that photothermal ablation could be used together with carbon nanotubes to trigger significant immune responses and, additionally, in combination with anti-CTLA-4 antibody therapy to prevent metastasis [165].

**2D nanomaterials**—the most popular 2D nanomaterials include those of the graphene family (graphene oxide (GO), reduced graphene oxide (rGO), graphene quantum dots, and graphene nanoribbons), black phosphorus (BP), layered double hydroxides (LDHs), transition-metal dichalcogenides (TMDs), transition-metal oxides (TMOs), and MXenes [166,167,168,169]. Biointeractions between the immune system and nanomaterials, including 2D nanomaterials, affect the immune system, and it is essential to identify all the affected factors that are related to biological safety. The cytotoxicity of graphene was studied, and this includes examining the effects on membrane integrity, cell viability and morphology, DNA damage, reactive oxygen species (ROS) generation, gene expression, and uptake mechanisms. The interactions of graphene oxide with cells can lead to excessive ROS generation, which is involved in the mechanisms of aging, carcinogenesis, and mutagenesis [170,171]. Graphene nanoparticles can disrupt the mitochondrial membrane potential and increase intracellular ROS generation, triggering the activation of apoptosis via the mitochondrial pathway [172]. Moreover, DNA intercalation and cleavage mechanisms are induced by the interactions of graphene with cellular genetic material [173]. Reduced graphene oxide (rGO) is a 2D nanomaterial of the graphene family with a single atom layer arranged in a honeycomb lattice structure. With a unique surface characterized by functional groups, rGO can be highly loaded with genes, increasing the delivery efficacy.

rGO was accepted for application in both chemotherapy combined with photothermal therapy and immunotherapy combined with photothermal therapy. A platform including PEGylated rGO combined with iron oxide was developed by Wang et al. [173,174]. Another strategy for targeting multiple antitumor immune pathways to induce synergistic antitumor immunity was explored in vivo by Yan et al., involving the design of a platform that combined immunotherapy with PTT based on folic acid and a multifunctional IDO inhibitor loaded with reduced graphene oxide (rGO)-based nanosheets (IDOi/rGO nanosheets), which directly kill tumor cells under laser irradiation and trigger an in situ antitumor immune response. IDO inhibition and the PD-L1 blockade triggered the immune response, enhancing tumor infiltration by lymphocytes, including T cells and NK cells, and suppressing Tregs and the production of IFN-γ in CT26 colon cancer cells [174].

Clinical research on nanopharmaceuticals for potentiating immunotherapy is currently at a point at which many have reached the clinical research phase, with promising results (Table 3).

### 4.2. Bioinspired Nanovectors

**Bacterial minicells** (Figure 3) are nanosized (100–300 nm in diameter) and originate from bacteria via abnormal cell division, and they have been used for the therapy of cancer, beginning several decades ago [185]. Minicells are nonliving, anucleated, nondividing cells that lack chromosomal DNA but contain RNA, ribosomes, peptidoglycan, proteins, and plasmids. Minicells still maintain other cellular activities, including mRNA translation, ATP synthesis, and the transcription and translation of plasmid DNA, but they have no possibility to grow or divide [186]. Some minicells have been used to deliver antigens, such as *Salmonella* (*S.*) *Typhimurium* T3SS, which was engineered into minicells. T3SS delivers antigens directly into the cytosol via the class I antigen presentation pathway (APC), stimulating antigen-specific CD8(^+^) T cells [187,188].

**Extracellular vesicles (EVs)** are a heterogeneous group of small membrane-bound vesicles that function as key mediators of many pathophysiological processes, with many advantages for drug and gene delivery and therapeutic capacities [189]. EVs are secreted by different cells, including dendritic cells (DCs), epithelial cells [190], neural cells [191], mesenchymal stem cells (MSCs) [192], and tumor cells (tumor-derived exosomes) [193]. EVs are also found in the serum, saliva, urine, and other bodily fluids [194]. Subtypes of EVs, including ectosomes, exosomes, microvesicles, membrane vesicles, and apoptotic bodies, have been described, which are isolated from different sources, and each type may have distinct biogenesis pathways, given that their biological origin is uncertain in most cases [195]. Proteomic evidence suggests that an EV core protein signature is commonly shared between EVs of diverse parent-cell origins [196,197]. EVs already have applicability in tumor immunotherapy due to possessing some advantages, such as lower toxicity and resulting in more frequent and durable responses.

**Exosomes** are a subtype of EVs that are secreted by the vast majority of cell types, with functions of intercellular communication and the transportation of proteins, lipids, and nucleic acids between cells and organs, thereby being involved in the progression of cancer. Exosomes are considered sophisticated vesicles involved in various physiological and pathological processes in the immune system based on their role as modulators, mediators, or activators.

A subgroup of exosomes, tumor-derived exosomes (TEXs), are used to load and deliver synthetic drugs, silencing RNAs and microRNAs. CD47v limits the ability of macrophages to devour tumor cells by binding to SIRPα, serving as a “do not eat me” signal [198]. Exosomes that carry SIRPα variants may antagonize the interaction between CD47 and SIRP, enhancing tumor phagocytosis and enhancing an effective antitumor T cell response.

The secretion of exosomes is a characteristic of both lymphoid and myeloid lineages, and also of many types of cells involving the TME and cancer cell secretion of tumor-derived exosomes that contain growth factors and microRNAs (e.g., miR-423-5p and miR-675) [199,200].

A combination of GM-CSF treatment and exosomes derived from ascites has demonstrated its potency in enhancing tumor-specific cytotoxic T cells, which determined the responses in a phase I trial [201]. The safety and therapeutic efficacy of autologous exosomes derived from dendritic cells carrying tumor MAGE peptides were demonstrated in another phase I trial, resulting in the improvement of antitumor immunity, and extending the disease responses in patients with advanced non-small-cell lung cancer (NSCLC) [202].

In a phase II clinical trial, exosomes loaded with IFN and MHC class I and II proteins were administered to patients with advanced NSCLC, resulting in an observed improvement in the NK cell-mediated antitumor immune response, with prolonged overall survival and progression-free survival in 50% of the patients [203]. Chimeric antigen (CAR)-T cell-derived exosomes also resulted in an improvement in clinical responses and benefits in controlling immune-related adverse events [204,205,206].

**Virus-like particles (VLPs)** were designed to mimic animal, plant, or bacterial viruses, but lack the property of replicating in human cells. VLPs are suited to the transport and delivery of nucleic acids due to their viral nature [207], but were developed as empty capsids that can also transport other types of molecules. VLPs have a structure that allows genetic and chemical engineering. It was demonstrated that nonhuman virus-based VLPs do not recognize human cell receptors and require genetic or chemical retargeting to malignant cells. The retargeting of VLPs using cancer-specific peptides, aptamers, or other molecules, or by enveloping VLPs with exogenous proteins, is possible. VLPs derived from human pathogens exhibit efficient gene transfer within human cancer cells. VLPs, being immunogenic, can enhance immune responses and can be engineered as vaccines to target immune cells. Lizotte et al. reported the use of a VLP-based vaccine based on cowpea mosaic virus as a delivery vehicle and also as an immunotherapeutic agent. VLPs specifically target TME cells and tumor cells and can be used as a nanocarrier for tumor antigens and drugs [208].

**Oncolytic viruses (OVs)** are viruses that have properties allowing them to infect and kill cancer cells. OVs are involved in the regression of tumors through target replication in tumor cells, promoting immunogenic cell death and host antitumor immunity. A large number of both DNA and RNA viruses have undergone preclinical studies as potential candidates for OV drug development. In OV selection, viruses with natural tropism and predilection for preferential replication in tumor cells are preferred, as are those demonstrating enhanced replication in tumor cells. The mechanisms of OVs involve direct oncolytic activity and the promotion of the immunogenic cell death of tumor cells, generating a tumor-reactive T cell response, especially for CD4^+^ and CD8^+^ T cells [209,210]. OVs can be used as platforms for the specific delivery of immune-stimulatory transgenes into the tumor niche (chemokines, cytokines, ICIs, co-stimulatory ligands, and tumor-associated antigens) to trigger immune-cell activation and transform the immunosuppressive TME [211]. Preclinical studies have demonstrated the use of engineered OVs in the delivery of a variety of genes that help in promoting the induction of immune responses, cytotoxic killing of tumor cells, suppression of tumor neoangiogenesis, radiosensitization, and other strategies [212,213]. For the delivery of the virus to patients with cancer, intratumoral (IT) injections were initially studied, but intravenous administration can also be considered, which allows the targeting of multiple metastatic lesions. 

In the last years, an abundance of clinical trials related to OV were published involving numerous viruses as strategies of treatment in various cancers. There are three oncolytic viruses approved for cancer therapy: Rigvir (a picornavirus) and Talimogene laherparepvec (a herpes simplex virus type 1) approved for the treatment of malignant melanoma, and a modified adenovirus H101 combined with cytotoxic chemotherapy for the treatment of nasopharyngeal carcinoma [131,133,214,215]. The most common oncolytic viruses that have been used for the treatment of cancer in clinical trials are adenovirus, HSV-1, reovirus, and poxviruses. The majority of studies used GM-CSG as a transgene for oncolytic viruses to enhance the recruitment and maturation of dendritic cells and, finally, to generate immune responses by promoting the cross-presentation of tumor antigens [216,217,218,219].

**Phage nanobioparticles (NBPs).** Phage display is a nanotechnology with potential and without any described limitations that was first developed in 1985. In 2018, the procedure for isolating high-affinity ligands for diverse substrates, ranging from recombinant proteins to cells, organs, and even whole organisms, was awarded the Nobel Prize in Chemistry. Its application in cancer immunotherapy involves three main types of phage-display procedures: (1) phage display-derived peptides that mimic cancer antigens (mimotopes), (2) antigen-carrying phage particles as prophylactic and/or therapeutic vaccines, and (3) phage display-derived peptides as small-molecule effectors of immune cell functions [220]. Some peptides displayed by phages can be used as modulators of immune system cells, stimulating effector cells (lymphocytes and APC) or inhibiting suppressor cells (Tregs and TAMs). The advantages of phages as carriers derive from their immunogenicity in association with low toxicity [221]. This technology has some limitations related to peptide stability and delivery, phage immunogenicity, and the challenges related to the diversity of human immunogenetics [222]. The technology of mimotopes of tumor-associated antigens (TAAs) that stimulate the production of anti-TAA Abs was developed, and these can be administered as full-length TAAs, partial proteins comprising only the antigenic parts, or TAA mimotopes (mimotopes of CD20, HER2, and EGFR). This type of cancer vaccination involves the presentation of a TAA or mimotope to the immune system of a patient to stimulate an immune response. Using mimotopes, this response may be focused against a peptide or against a peptide genetically or chemically anchored to the phage surface. It was demonstrated that the aggregation of a peptide with a phage particle results in a better response and a lower immunogenicity in comparison to the use of another carrier, e.g., ovalbumin (OVA) [222]. Another application of phage display is carrying derived peptides as nanomodulators of the immune response. One of the procedures of phage display involves the indirect targeting of the immune system. Phage display-derived peptides functioning as nanomodulators of the immune response with two main applications have been described, namely, in interfering with the activity of immune cells or with immune checkpoints.

## 5. Potential Targets for Nanomedicine-Based Cancer Immunotherapy

The aims for using nanovectors in cancer are to improve the delivery of therapeutic drugs to tumors and metastases and to modulate the immune system, for which three principal targets exist and represent directions for further exploration: cancer cells, the tumor immune microenvironment (TIME), and the peripheral immune system. Various types of inorganic nanoparticles have been designed for targeting cancer cells, the TIME, and the peripheral immune system, including PLGA, liposomes, hafnium oxide NPs, gold NPs, micelles, mesoporous silica, and carbon nanotubes. Current cancer immunotherapies are often based on the use of ACT, therapeutic cancer vaccines, and monoclonal antibodies.

### 5.1. Cancer Cells as a Target for NP-Based Immunotherapy

Nanovectors can be used to enhance the induction of immunogenic cell death (ICD). ICD can be induced by chemotherapeutics (e.g., doxorubicin, oxaliplatin, and cyclophosphamide) or radiotherapy, magnetic fluid hyperthermia, photodynamic therapy, or other stimuli [223]. So-called “in situ tumor vaccines” are nanoparticles designed for ICD that provide a new way to promote more efficient immunotherapy through combination with ICD-inducing modalities. ICD is characterized by the release of TAAs and danger-associated molecular patterns (DMAPs). ICD is characterized by the translocation of calreticulin (CRT) to the cell surface and release of ATP and the high mobility group box 1 protein (HMGB1) into the extracellular environment. These modifications alert the immune system, which activates APC and cytotoxic T cells that eradicate tumors and metastases. Doxorubicin-loaded liposomes (Caelyx/Doxil) can increase the efficacy of immunotherapy when combined [224]. It is supposed that Doxil, through ICD, promotes the proliferation of DCs and CD8^+^ T cells. It was demonstrated that the immunopotentiation for Doxil is higher than that for doxorubicine administered at the same dose. Similar results were reported for oxaliplatin-loaded PLGA nanoparticles, which induce ICD and are more efficient in activating the immune system than free oxaliplatin [225].

For tumor-targeted delivery, immunotherapeutic agents were also used in combined photodynamic therapy–radiotherapy. It was demonstrated that pyrolipid-loaded inorganic nanoparticles enhance immunoactivation and ICD induction in photodynamic therapy when combined with anti-PD-L1. This ICD induction increases the serum levels of proinflammatory cytokines, such as TNF-α, IL-6, and IFN-γ, while also improving the tumor infiltration of CD4^+^ and CD8^+^ cells, eradicating the primary tumor, and preventing lung metastasis through an abscopal effect [226]. The abscopal effect represents a phenomenon whereby local radiotherapy induces a systemic immune response and the regression of metastatic lesions [227]. Blocking TGF-β activity during radiation therapy was observed to generate CD8^+^ T cell responses to endogenous tumor antigens. The addition of anti-PD-1 and/or anti-CD137 antibodies resulted in a prolongation of survival achieved with radiation in combination with the TGF-β blockade [228,229,230,231]. Nanomedicines inducing ICD result in improved antitumor immunity through decreased systemic lymphocyte toxicity, which also potentiates immunotherapy outcomes [232]. Moreover, locally injected nanoparticles are able to induce systemic immunity via an abscopal effect [233]. The European Medicines Agency approved intratumoral injections of NBTXR3 hafnium oxide nanoparticles designed to enhance the radiation-induced abscopal effect of radiotherapy for patients with locally advanced soft-tissue sarcomas [234].

### 5.2. Tumor Immune Microenvironment (TIME) as a Target for NP-Based Immunotherapy

Immunosuppressive mediators and pathways are upregulated in the TIME by an increase in the infiltration of immunosuppressive cells, such as tumor-associated macrophages (TAMs) and MDSCs, into tumors and enhancement of the levels of inhibitors, such as IDO and TGF-β.

Moreover, tumor acidity and hypoxia are the two main conditions that can be targeted to modify the immune response. It has been demonstrated that hypoxia has a negative regulatory effect on T cell activation, enhancing the expression of CCL22 and CCL28 and resulting in MDSC and Treg accumulation [235].

Hypoxia increases the secretion of immunosuppressive factors (VEGF and TGF-β) in addition to the expression of PDL-1 on the T cell immunoglobulin domain, mucin domain-3 (TIM-3), and CTLA4 on MDSCs, TAMs, and Tregs [236]. pH/H_2_O_2_ dual-responsive nanoparticles were designed using albumin-coated MnO_2_. Upon tumor penetration, MnO_2_ reacted with H_2_O_2_ and H^+^ to produce oxygen, enhancing the therapeutic effects of chemotherapy and photodynamic therapy [237]. Another immunoliposome was developed, known as CAT@aPDL1-SSL, which represents another immunoliposome developed that contains catalase (CAT)-encapsulated liposomes and a modified PDL-1 for improving immunotherapeutic effects, enhancing T cells in tumor tissues, and blocking the PD-1/PD-L1 pathway. The liposomes decrease hypoxia through the decomposition of endogenous H_2_O into O_2_ [238]. The extracellular pH of most tumor tissues is found to be lower than that of normal tissues [239]. The mild acidic tumor microenvironment is the consequence of the high glycolysis rate of tumor cells, with lactic acid accumulation.

pH-responsive PCL–Hyd–PEG nanovesicles that encapsulate immunological adjuvants (CpG ODNs) and endogenous tumor antigens as heat shock protein 70-chaperoned polypeptides (HCP) were constructed to enhance cancer immunotherapy. These nanovesicles are fragmented in the acidic tumor microenvironment and release the encapsulated drugs [240]. Furthermore, pH-responsive size-switchable or dissociable nanoparticles were developed. It was demonstrated that nanoparticles with smaller diameters efficiently expanded into the tumor, and N-(2-hydroxypropyl) methacrylamide (HPMA) polymer-based nanovehicles with a small size and ability to change the pH were developed for the delivery of chemotherapeutic drugs to the nucleus [241].

**Dendritic cells (DCs)****as a target for NP-based immunotherapy.** Dendritic cells (DCs) are multifunctional regulators of immunity. 

It has been well demonstrated a cross-presentation process of antigens endocytosed by DCs to CD8^+^ cytotoxic cells [242]. Tumor-associated antigens and adjuvants targeting the dendritic cells and tumor-specific T cells were used as therapeutic cancer vaccines and positive results were reported in preclinical and clinical experiments involving immunotherapy with manipulated DCs [243]. Sipuleucel-T was the first DC-based vaccine approved by FDA for patients with hormone-resistant advanced prostate cancer. Other strategies targeting DCs have been experimented *in vivo* [244]. The selective uptake of DCs and the decreasing of off-target drug interactions have been also studied [245,246,247].

One application of DC targeting by nanomedicines is in RNA modulation. In a study on specific delivery systems, siRNA carried by liposomes was targeted to DCs for silencing CD40 expression in vitro [248].

In the clinical testing of vaccines, low-immunogenicity lipid-based RNA nanoparticles were designed for the delivery of mRNA into DCs. These widely used cationic lipid materials (DOTMA, DOTAP, and DOPE) and anionic mRNA form RNA lipoplexes that ensure efficient and precise DC-targeted mRNA delivery without the need for molecular ligands, such as antibodies [249]. Nanoparticle-mediated hyperthermia activates DCs, and combination therapy with magnetic nanoparticle-induced hyperthermia, radiotherapy, and a virus-like particle adjuvant was demonstrated to be effective in the treatment of dogs with oral melanoma [250].

**Tumor-associated macrophages (TAMs) as a target for NP-based immunotherapy.** TAMs are immune cells with an M2-like phenotype in tumors with pro-tumoral functions in suppressing the infiltration of effector T cells [251,252]. It was demonstrated that ferumoxytol changed M2-like TAMs into M1-like TAMs and inhibited the growth of primary and metastatic tumors in the liver and lungs [253]. In another study, it was found that cyclodextrin nanoparticles are targeting a small-molecule Toll-like receptor 7/8 agonist that binds TLR7/8 intracellular receptors expressed on the endosomal membranes of macrophages in the TIME, resulting in the induction of M2–to–M1 polarization, enhancing the efficacy of checkpoint inhibitors [254].

The increase in macrophages polarized toward an M1 phenotype is followed by improved outcomes of checkpoint–blockade therapy when using CaCO_3_ nanoparticles functionalized with anti-CD47 antibodies [255]. Two immunosuppressive molecules from the TIME—IDO and TGF-β—were also targeted by the nanoparticles. 

**Indoleamine 2,3-dioxygenase (IDO) as a target for NP-based immunotherapy.** The role of IDO is to enhance the production of kynurenine, a T cell-suppressing metabolite. Molecules of IDO inhibitors incorporated into nanomedicine formulations were tested in preclinical and clinical trials, and synergic mechanisms between IDO inhibitor-loaded nanomedicines and photodynamic therapy and radiotherapy were observed [256]. An IDO inhibitor was combined with oxaliplatin to enhance ICD in lipid-coated mesoporous silica nanoparticles, followed by tumor reduction in pancreatic ductal adenocarcinoma [257]. Another IDO inhibitor was used together with a peptide that blocked PD-L1 in peptide-based nanoparticles, which effectively inhibited melanoma growth in mice by stimulating anticancer immunity [258].

**TGF-β****as a target for NP-based immunotherapy.** TGF-β was found to inhibit the efficacy of checkpoint inhibitors [259,260]. A TGF-β inhibitor encapsulated in PEGylated immune liposomes was demonstrated to increase the T cell triggering of the receptors CD90 and CD45 [261]. TGF-β-siRNA-containing nanoparticles were developed, which were shown to downregulate TGF-β expression in tumors [262].

### 5.3. Peripheral Immune System as a Target for NP-Based Immunotherapy

The peripheral immune system, defined as immune compartments located outside tumors, is mainly composed of secondary lymphoid organs, which are the places where antigen presentation and cytotoxic-T cell generation occur. These compartments are frequently impaired in cancer progression. The functions of the peripheral immune system can be restored by the potentiation of antigen presentation and by engineering T cells. The subcutaneous or intradermal administration of antigen-containing nanoparticles results in more efficient processing by APCs [263]. CpG conjugated with nanoparticles or loaded together with peptide antigens in nanodiscs was administered in local injections targeting lymph nodes for promoting anticancer immunity [264,265]. In addition, the Toll-like receptor 7/8 agonist imidazoquinoline entrapped in nanogels or CpG bound with albumin was injected locally or systemically with the intention of it reaching the lymph nodes. Such vaccines demonstrate the tolerability of adjuvants [266].

Another antitumor vaccine was developed using PLGA nanoparticles containing antigens, which were administered to target lymph nodes for the delivery of antigens to DCs, resulting in significantly improved immunotherapy and an ex vivo abscopal effect in tumor-bearing mice receiving αPD-1 immunotherapy treatment [267,268]. A nanovaccine was designed based on the combination of an antigen and a synthetic polymeric nanoparticle, PC7A-NP, which, after administration, delivered antigens to antigen-presenting cells from lymph nodes, activating type I interferons. This vaccine, in combination with the anti-PD-1 antibody, resulted in 100% survival over 60 days when applied in a tumor model [269]. Another strategy involves generating cytotoxic T cells to replace APCs. Synthetic APCs were designed based on a polypeptide modified with anti-CD3 antibodies included in the polymer chain, which enhanced the expression of CD69 [270]. Synthetic biometric magnetosomes were also prepared as versatile artificial APCs that trigger cytotoxic T cells, and they promote tumor inhibition when administered together with T cells in tumors [271]. Liposomes loaded with IL-15 and IL-21 or cytokine-based nanogels modulating the release of IL-15 were also studied [272]. An antigen-encoding mRNA within a lipoplex has already been studied in clinical trials, as a monotherapy or combined with immunotherapeutics [273].

## 6. A Possible Mathematical Model for Cellular Communication Mechanisms

All cells in multicellular beings have the distinct requirement of communication with other cells for coordinating development and for adaptation and functional evolution. This communication involves many types of soluble factors in addition to specific recognition through cell-surface receptors. There is recent evidence that cells communicate directly via RNA exchange. When eukaryotic cells encounter double-stranded RNA (dsRNA), genes carrying a matching sequence can be silenced through RNA interference (RNAi).

However, the novelty is that, in some animals and plants, transporting a silencing signal between cells will allow the same gene to be specifically silenced in cells that had not encountered the primary dsRNA. This process can be seen in plants and *C. elegans*. In plants, silencing RNAs move from cell to cell through the plasmodesmata (PD), and over long distances through the phloem vascular tissue. When a leaf is, in general, infected with a virus, such mobile signals transmitted to others provide resistance to infection spread. Even if it was known for years that transgenes and viral-induced siRNAs move through the plant, the movement of endogenous small RNAs (sRNAs) has only recently been demonstrated.

The movement of endogenous small RNAs, which includes microRNAs (miRNAs), leads to signal gradients that may guide the patterning of leaves and roots. Such mobile sRNAs can promote epigenetic modifications in the genomes of recipient cells. Furthermore, if such recipient cells are pollen or seed-mobile, the sRNAs can induce transgenerational epigenetic changes, which enhance progeny adaptation to future stresses. In C. elegans, silencing initiated by dsRNA spreads through the organism, which silences the targeted gene in all non-neuronal cells, including the germline, thus transmitting this silencing to the next generation.

Intercellular communication has been extensively studied, since it helps to directly transfer information between cells or through various molecules created by them. Extracellular vesicle (EV) secretion is a common process that has been identified in many biological fluids. EVs facilitate the exchange of nucleic acids, lipids, and proteins between cells, playing an important role in cell signaling [274].

Cell communication in the human body and, especially, in cell microenvironments plays an essential role in cancer development and tumor growth [275]. Exosomes are a class of EVs secreted by all types of cells that are involved in intracellular communication with other nearby or distant cells, immunological actions, cancer metastasis, or other organ-specific processes [276]. They are composed of a phospholipid double layer and are between 50 and 100 nm in diameter [277]. They are important in all components of a cell, even in the case of DNA, miRNA, mRNA, or proteins. Although exosomes were first described 50 years ago, their role in the development of cancer as potential “biomarkers” has been extensively researched over the past decade [278].

The classical models used in these studies are usually based on the unjustified assumption that the variables describing the dynamics of any cell complex are differentiable [279]. Therefore, the success of these models must be understood as sequential, existing only in areas where differentiability and integrability are valid. These procedures are not appropriate when the dynamics of a cell complex involve nonlinearity (chaoticity and self-structuring).

However, the notion of scale resolution for variable expression must be introduced to describe these dynamics using differential and non-differential procedures, especially in the expression of fundamental equations that control such dynamics. Thus, any variable that depends on spatio-temporal coordinates will depend on space and time, yet also scale resolutions in the new mathematical sense, which is that of non-differentiability and non-integrability. 

The functionality of a variable described by non-differentiable functions is replaced with approximations of this function obtained through averaging at various scale resolutions. Resulting from these procedures, any and all variables developed in order to describe complex cellular dynamics can be a limit of function families that are nondifferentiable for a zero-scale resolution and differentiable for non-zero-scale resolutions [280,281].

The method of describing the dynamics of any complex cellular system involves the development of new geometric structures in addition to new mathematical models. For these, the laws of motion, invariant to spatial and temporal transformations, are integrated with scaling laws, which are also invariant. According to the authors, such a structure can be based on the concept of multifractality, and the corresponding model can be based on the fractal theory of motion in an arbitrary and constant fractal dimension. For complex biological systems, the dynamics analysis is similar to the one given in [282].

The fundamental assumption of the model is that the dynamics of the structural units of any complex cellular system will be described by continuous and nondifferentiable curves (motion multifractality curves). They have properties of self-similarity at each point, which can be translated as properties of holography (where each part reflects the whole). Basically, the discussion revolves around “holographic implementations of the structural unit dynamics of any cellular complex system” through multifractal “regimes” [282]. 

This implementation naturally implies many types of operational procedures (invariance groups, harmonic mappings, groups isomorphism, embedding manifolds, etc.) with many applications in complex systems dynamics, such as the matter of explaining the dynamics of a cellular system. It is thus possible to observe and quantify nonlinear behaviors at a global scale resolution, nondissipative nonlinear behaviors at a differentiable scale resolution, and even dissipative nonlinear behaviors at a nondifferentiable scale resolution. Finally, whatever the type and exact measure of the scale resolution, such cellular dynamics can be reduced to self-structuring cellular patterns that, through the phasing and dephasing of the positive and negative parts of their respective complex potential fields, can dictate where and how EV secretion can take place.

## 7. Discussion

Nanopharmaceuticals and bioinspired nanovectors (nanobioparticles) remain the two main categories of nanovectors involved in the modulation of cancer immunotherapy. However, great efforts are still required to facilitate further understanding of the in vivo fate of nanopharmaceuticals.

A new research strategy for influencing immunotherapy is that represented by 2D nanomaterials. Two-dimensional (2D) nanomaterials differ from their conventional zero-dimensional (0D) and one-dimensional counterparts in showing unique properties that result from their specific structure and morphology. Two-dimensional-based drug-delivery systems show great potential and represent a novel direction for the expansion of innovative biomedical applications. 

Biointeractions between the immune system and nanomaterials, including 2D nanomaterials, affect the immune system, and it is essential to identify all the factors related to biological safety. 

The complexity and heterogeneity of the immune–nanomaterial interface still represent an issue in developing nanopharmaceuticals. Different interactions between nanomaterials (NMs) and immune cell membranes result in different immune responses. The immune–nanomaterial interface is represented by specific characteristics, such as stereoselective interaction, hydrophobicity, electrostatic interaction, hydrogen bonding, metal coordination, and molecular recognition, which cause structural remodeling and the dysfunction of nanomaterials. Consequently, the surface properties, biological functions, intracellular-uptake pathways, and fate of nanomaterials are affected [283].

The existence of entropic and nanothermodynamic potentials at immune–nanomaterial interfaces have also been described, regulating the receptors on macrophage surfaces and stimulating the secretion of cytokines [284].

At present, there is a sustained effort toward developing comprehensive databases, for example, protein corona fingerprints and a biological response database of various types of NMs and a nanocombinatorial library strategy [285,286].

The development of databases for nanomaterial-related immunity is still in its infancy, but these databases can predict biological responses and the basic mechanisms of nanoimmune interactions, and allow screening for safe and effective NMs. Moreover, the combination of multivariate immune analysis, data integration, and machine learning can provide insight into the impact of the immune responses induced by nanomaterials [287,288].

Another route in the development of nanovectors is represented by bioinspired vectors, with major potential for development in the coming decades. New strategies to improve oncolytic virotherapy in the future have been described.

A strategy based on the use of a protective coating to physically shield OVs to prevent the action of soluble immune factors was developed, involving the use of liposomes, polymers, and cell-derived nanovesicles [289]. Cellular carriers derived from T cells, endothelial cells, mesenchymal stromal cells, and also from adipose-derived stem cells were designed to deliver viruses, such as vaccinia virus and vesicular stomatitis virus [290,291]. Some epitopes on OVs were also identified and genetically modified, preventing the premature exposure of pre-existing antibodies [292,293].

## 8. Conclusions

Future directions involving nanomedicine to modulate immunotherapy in cancer treatment are needed to increase the effectiveness of treatments. The major issues in nanotechnology-based immunotherapy are represented by the optimization of tumor targeting, the control of toxicity, and drug delivery versus clearance.

Additionally, complementing classic checkpoint inhibitors is the promising discovery of new checkpoint co-stimulators, such as OX40/OX40L, which promotes the survival and proliferation of CD4 and CD8 T cells; GITR/GITRL, which exerts a regulatory function on Tregs; and 4-1BB/4-1BBL, which has a co-stimulatory effect on different immune cells (T cells, NK cells, Tregs, and NK T cells). Checkpoint inhibitors, such as LAG-3, TIGIT, VISTA, and TIM-3, or enhancers of cellular immunity (STING agonists, IDO/TDO, and TRL agonists) are also of interest.

The major direction for developing nanovector technologies to enhance immunotherapy appears to involve nanopharmaceuticals, bioinspired nanoparticles, and combinations of them, which have great potential to be demonstrated in the coming decades.

## Figures and Tables

**Figure 1 pharmaceutics-14-00397-f001:**
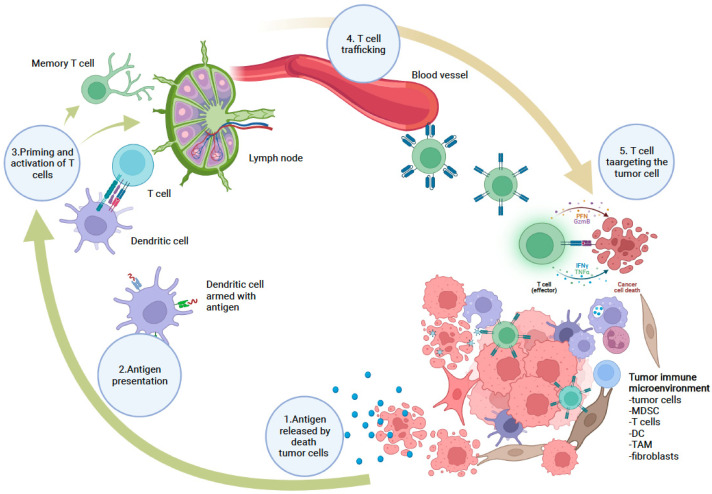
Process of antigen release by the tumor cell is followed by processing and presentation by APCs and activation of effective immune cells. T cells are trafficking and infiltrating the tumor tissues, being activated the immune cells from the TIME (tumor-infiltrating microenvironment) (created with www.BioRender.com).

**Figure 2 pharmaceutics-14-00397-f002:**
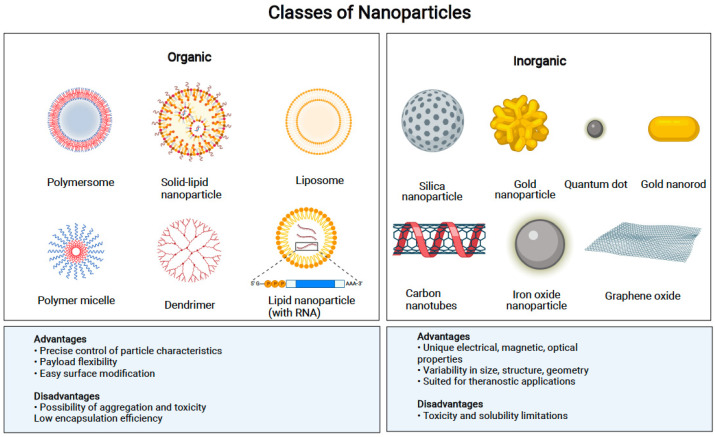
Classes of nanopharmaceuticals (created with BioRender—www.BioRender.com).

**Figure 3 pharmaceutics-14-00397-f003:**
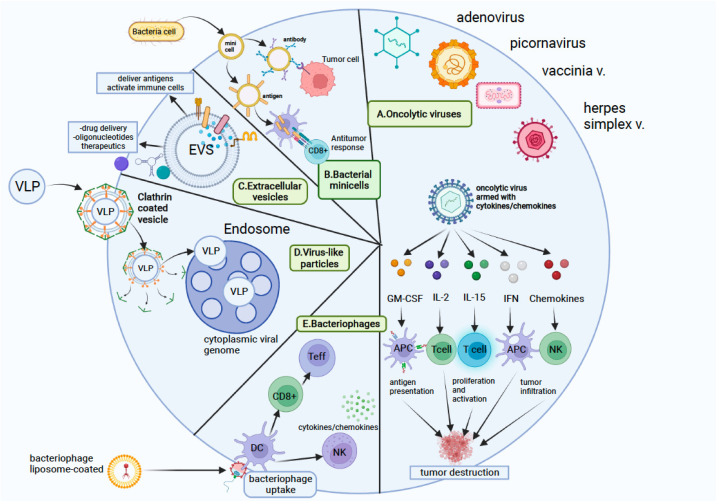
Bio-inspired nanovectors: A. Tumor destruction by cytokine/chemokine-armed oncolytic viruses. GM-CSF helps in antigen presentation through the recruitment and activation of dendritic cells and macrophages; B. Minicells with bacterial origin encapsulate the chemotherapeutic drugs, shRNA, or toxin protein. Minicells bind to the receptors of the cancer cells via the bi-specific antibody conjugated on the minicell’s surface. Minicells enter the cancer cell, where they release drugs with an anti-cancer effect. Engineered minicells can produce antigen, which activates the anti-tumor immune response; C. EVS deliver antigens to activate the immune cells or can deliver anti-tumor drugs D. VPL are attached to host receptors, inducing the binding of adaptor proteins to clathrin and releasing of the clathrin-mediated vesicles that deliver to early endosomes; E. Bacteriophages can be encapsulated into polymers and stimulate the DCs, triggering an immune response (created with Biorender—www.BioRender.com).

**Table 1 pharmaceutics-14-00397-t001:** Examples of the nanopharmaceuticals used for immunomodulation in cancer therapy.

Nanovector Type	Nanovector Family	Delivery Platforms	Mechanism of Action	Types of Cancer	References
Organic nanoparticles	PLGA	PLGA transporting TLR7/8 bi-specific agonists	Increased co-stimulatory molecule expression and antigen presentation via MHC I by DCs	Melanoma, bladder, and renal-cell carcinoma	[104]
		PLGA carrying siRNA and R837	PLGA NPs with RNA (siRNA) for knockdown of STAT3 in DCs and imiquimod, R837 for activating DCs through TLR7		[105]
		PLGA-NP carrying murine melanoma antigenic peptides, hgp100(25-33) and TRP2(180-188)	Increased MHC class I expression and enhanced tumor control, DC maturation and activation		[106]
	Dendrimers	2G-03NN24 dendrimer	Decreasing expression of M2-polarization genes, decreased STAT3 activation		[107]
	Liposomes	MgluPG + pDNA liposome complexes (lipoplex)	Transfecting DC2.4 cells and inducing IFN-γ protein production		[108]
		Liposomal encapsulated agonists of STING	Improving the cellular uptake of cGAMP and proinflammatory gene induction	Melanoma, lung	[109]
		PEGylated YSK05-MEND	Gene silencing	Subcutaneous tumor	[110]
		Liposome–protamine–hyaluronic acid (LPH) NP + siRNA	Knockdown of TGF-β	Melanoma	[111]
	Micelles	Polymeric hybrid micelles (PHMs) with Trp2/PHM/CpG co-delivery system	Enhance antigen-specific cytotoxic T-lymphocyte activity	Melanoma	[112]
		Galactose-functionalized zinc protoporphyrin IX (ZnPP) grafted poly(l-lysine)-b-poly(ethylene glycol) polypeptide micelles (ZnPP PM)	Repolarization of TAMs to antitumor M1 macrophages		[113]
Inorganic nanoparticles	Gold nanoparticles	CpG oligodeoxynucleotide + GNPs	- Delivering CgP oligonucleotides - Promoting infiltration of macrophages and DCs		[114]
		GNPs + model antigen (BSA) CpG oligodeoxynucleotides	Activating the immune response of macrophages by interacting with TLR9 receptor		[115]
	Iron oxide NPs	Fe_3_O_4_−OVA nanoparticle vaccine	Promoting secretion TNF-α, IL-6, and IFN-γ	Colon	[116]
	Mesoporous silica (MSNPs)	XLMSNs + OVA + CpG-ODN vaccine	Inducing DC maturation, enhancing IL-12 and TNF-α		[117]
		MSNPs + indoximod + oxaliplatin	- IDO inhibition - Induction of immunogenic cell death	Pancreas	[118]
	Carbon nanotubes (MWNTs)	(αCD40)S ± (OVA−CpG) incorporated in MWNTs	Enhancement of OVA delivering specific immune response	Melanoma	[119]
	Carbon nanotubes	CNT-loaded Rg3	- Suppress the PD-1/PD-L1 axis - Enhance the levels of IFN-γ and interleukins-2, 9, 10, 22, and 23	Triple-negative breast cancer	[120]
	Graphene oxide (GO)	Graphene quantum dots (GQDs)	Inducing apoptosis, autophagy, and inflammatory response in activated THP-1 macrophages		[121]
	Reduced GO) (rGO)	PEG–rGO–FA–IDOi	IDO inhibition and PD-L1 blockade that enhances TILs and suppress Tregs		[122]
		(IDOi/rGO nanosheets)			

**Table 2 pharmaceutics-14-00397-t002:** Examples of the bioinspired nanovectors used for immunomodulation in cancer therapy.

Nanovector Type	Nanovector Family	Platform	Mechanism	Types of Cancer	References
Bacterial minicells	Minicells	*Salmonella* (*S.*) *Typhimurium* T3SS	T3SS deliver APC and stimulate CD8^+^ T cells		[123]
Extracellular vesicles	Tumor-derived exosome (TEXs)	SAV-exo + CpG-SAV-exo	- Strong Th-1 antigen-specific immune response - Strong tumor-specific CD4þ and CD8þ T cell responses	Melanoma	[124,125]
		Adenovirus platform (LOAd) with transgenes (TMZ–CD40L and 4 1BBL)	Targeting DCs, T cells, and NK cells	Melanoma	[126]
	Dendritic cell-derived exosomes (DEX)	DEX + MAGE tumor antigen	- Activating antigen-specific, MHC-restricted T cells - Directly activating NK cells	NSCLC	[127]
		DEX + IFN-γ	Activating NK cells		[128]
	Ascites-derived exosomes (Aex)	Aex + GM-CSF	- CEA (CAP-1 peptide)-specific IFN-γ release from CD8^+^ T lymphocytes - Promoting antigen presentation and T cell activation	Colon	[129]
Virus-like nanoparticles (VLNs)	VLNs	VLN-sgPD-L1@Axi	- Co-delivery system to enhance efficacy of CRISPR/Cas9 - Disruption of PD-1/PD-L1 pathway - Reinvigoration of T cells and TILs		[130]
Oncolytic viruses	Picornavirus	ECHO-7 strain of a picornavirus	Selectively infecting and destroying cancer cells	Melanoma	[131]
	Adenovirus	Engineered adenovirus H101	Interacting with normal human gene p53	Nasopharyngeal	[132]
	Herpes simplex virus	Herpes simplex virus encoding GM-CSF	- GM-CSF genes, replacing virulent ICP47 genes, stimulating CD8^+^ cells, accumulation of DC - Activating JAK–STAT pathways, stimulating IFN production - Inhibiting Tregs and MDSCs - Stimulating the production of - anti-Melan A/IFN-γ T cells	Melanoma	[133]
	Adenovirus (Onc.Ad)	CAR-T cells + Cad-VECPDL1	- CAR-T cells—produce proinflammatory cytokines - Onc.Ad—direct cytolysis of tumor cells - Local production of mini-body PD-L1 at the tumor site	Prostate	[134]
Bacteriophages	Bacteriophages	l ZAP-CMV-apoptin recombinant NBP	Transfection of the human breast neoplastic cells with the nanobioparticles carrying l ZAPCMV-apoptin construct		[135]
		Lambda-phage nanobioparticles containing enhanced EGFP and E7 gene of HPV type 16	Gene delivery system and vaccine recombinant lambda bacteriophages for gene delivery		[136]
		TAA-mimic molecule (mimotope)	The mimotope triggers the production of anti-TAA Abs		[137]

**Table 3 pharmaceutics-14-00397-t003:** Examples of ongoing clinical trials using nanovectors to enhance immunotherapy.

Nanovectors	Indications	Clinical Stage	Ref. Number www.ClinicalTrials.gov	Reference
PEGylated IL-2 + checkpoint inhibitor	Solid tumors	Phase I–III	NCT02983045 NCT03282344 NCT03785925 NCT03729245 NCT03435640	[175,176,177]
NBTXR3 activated by radiotherapy + anti-PD-L1	Solid tumors	Phase I	NCT03589339	[178,179]
Polymer with undisclosed payload nanoparticles + chemotherapy	Solid tumors	Phase I	NCT03781362 NCT03953742	[180,181,182]
Metallic–organic nanoparticles + IDO inhibitor to enhance radiotherapy ± checkpoint inhibitors	Solid tumors	Phase I	NCT03444714	[183]
Lipid nanoparticles to deliver mRNA encoding OX-40L	Solid tumors/lymphoma	Phase I/II	NCT03323398	[184]

## Data Availability

All data are presented in the manuscript.

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
