# Peer review of "The Landscape of Nanovectors for Modulation in Cancer Immunotherapy"

_pharmaceutics, 2022, doi:10.3390/pharmaceutics14020397_

Round 1

Reviewer 1 Report

In this review, different types of cancer immunotherapy are reviewed, and the research progress of different nanoparticles in tumor immunotherapy is summarized. In order to improve the depth of the discussion, the author should answer the following questions:

  1. Authors should add some schematic diagrams to representative papers, such as the action mechanism of different nanoparticles.
  2. The author enumerates the role of different nanoparticles in tumor immunotherapy. The author should also describe the shortcomings of these nanoparticles and the researchers' solutions to these shortcomings.
  3. The formation of the nanoparticle protein crown may cause the failure of immunotherapy. How to deal with this phenomenon? What is the future development direction of nanoparticle immunotherapy?
  4. The author said “2D-based drug delivery systems provide great potential for tumor therapy”, such as graphene oxide (GO), reduced graphene oxide, (rGO), black phosphorus (BP), layered double hydroxides (LDHs), etc. What is the biosafety of these materials, how do they get metabolized when they come into the body? Is there any possibility of clinical transformation?

Author Response

To: Reviewer1

Thank you very much for your suggestions. Our answers are the following (in yellow):

In this review, different types of cancer immunotherapy are reviewed, and the research progress of different nanoparticles in tumor immunotherapy is summarized. In order to improve the depth of the discussion, the author should answer the following questions:

  1. Authors should add some schematic diagrams to representative papers, such as the action mechanism of different nanoparticles.

We have added

  1. The author enumerates the role of different nanoparticles in tumor immunotherapy. The author should also describe the shortcomings of these nanoparticles and the researchers' solutions to these shortcomings.

We have added and described the shortcomings

  1. The formation of the nanoparticle protein crown may cause the failure of immunotherapy. How to deal with this phenomenon? What is the future development direction of nanoparticle immunotherapy?

We have changed and added these information

  1. The author said “2D-based drug delivery systems provide great potential for tumor therapy”, such as graphene oxide (GO), reduced graphene oxide, (rGO), black phosphorus (BP), layered double hydroxides (LDHs), etc. What is the biosafety of these materials, how do they get metabolized when they come into the body? Is there any possibility of clinical transformation?

We have change the 2-D materials description from Conclusions to Nano-pharmaceutilcals chapte and we added data about the toxicity and the pharmacokinetics of graphene family

We made major revision to the article. Also, we have corrected the English spelling errors (English-editing-certificate attached).

We have attached the revised article.

Sincerely yours,

Constantin Volovat

Reviewer 2 Report

The manuscript entitled “The landscape of Nanovectors that modulate the Immunotherapy in Cancer " is an interesting review work that assesses the potential of nanotechnology, and particularly nanovectors to improve the efficacy of immunotherapeutics in cancer.  The authors present a well-structured manuscript, however some information needs to be added for a better understanding. Thus, some comments are listed below:

1-Abstract is confused and does not reflect the structure and themes covered in the manuscript. You should better relate the contents described throughout the manuscript.

2-Lines 90-91 – “These tumors that are selective controlled by parts of the immune system.”. There is some missing information to complete the idea of the sentence.

3-Paragraph starting at line 98 – you correctly defined the three main phenotypes of TME, however I suggest adding some tumor examples for each phenotype for a better understanding.

4-Paragraph starting at line 139 – “Section Bioactive Nanoparticles designed to modulate cancer immunotherapy” – you should describe the most important physical and chemical properties of NP and how they influence the biological and immune characterization of an appropriate nanovector to modulate tumor microenvironment. These definitions are more important than given various examples of nanovectors.

 5-Lines 232-233 “Other pH-sensitive liposomes are curdlan and mannan used as bioactive polysaccharides, that deliver antigenic proteins into the DCs [40].” Please reformulate this sentence to “Curdlan and mannan are bioactive polysaccharides that can be used in the formulation of pH-sensitive liposomes to improve the activation of DC.” This sentence is one of many poorly structured found in the manuscript.

 6- Some information regarding the existence of nanovectors alone or in combination with immune checkpoint inhibitors (for example) in clinical trial studies should be added. This information demonstrates the potential of nanotechnology to improve existing immunotherapies.

 7-This manuscript does not present any image for an immediate understanding of some contents. You should prepare and add some images to the manuscript to improve its value.

8-Please extensively review all manuscript and pay attention to many grammatical and punctuation errors. Spelling should be improved. Tables should be also reviewed.

Author Response

Reviewer2

Thank you very much for your suggestions. Our answers are the following (in yellow):

Comments and Suggestions for Authors

The manuscript entitled “The landscape of Nanovectors that modulate the Immunotherapy in Cancer " is an interesting review work that assesses the potential of nanotechnology, and particularly nanovectors to improve the efficacy of immunotherapeutics in cancer.  The authors present a well-structured manuscript, however some information needs to be added for a better understanding. Thus, some comments are listed below:

1-Abstract is confused and does not reflect the structure and themes covered in the manuscript. You should better relate the contents described throughout the manuscript.

We have modified the abstract

2-Lines 90-91 – “These tumors that are selective controlled by parts of the immune system.”. There is some missing information to complete the idea of the sentence.

We have corrected

3-Paragraph starting at line 98 – you correctly defined the three main phenotypes of TME, however I suggest adding some tumor examples for each phenotype for a better understanding.

We have added

4-Paragraph starting at line 139 – “Section Bioactive Nanoparticles designed to modulate cancer immunotherapy” – you should describe the most important physical and chemical properties of NP and how they influence the biological and immune characterization of an appropriate nanovector to modulate tumor microenvironment. These definitions are more important than given various examples of nanovectors.

We have added

 5-Lines 232-233 “Other pH-sensitive liposomes are curdlan and mannan used as bioactive polysaccharides, that deliver antigenic proteins into the DCs [40].” Please reformulate this sentence to “Curdlan and mannan are bioactive polysaccharides that can be used in the formulation of pH-sensitive liposomes to improve the activation of DC.” This sentence is one of many poorly structured found in the manuscript.

We have reformulated.

 6- Some information regarding the existence of nanovectors alone or in combination with immune checkpoint inhibitors (for example) in clinical trial studies should be added. This information demonstrates the potential of nanotechnology to improve existing immunotherapies.

We have added some examples of trials

 7-This manuscript does not present any image for an immediate understanding of some contents. You should prepare and add some images to the manuscript to improve its value.

We have added

8-Please extensively review all manuscript and pay attention to many grammatical and punctuation errors. Spelling should be improved. Tables should be also reviewed.

We have revised

We made major revision to the article. Also, we have corrected the English spelling errors (English-editing-certificate attached).

We have attached the revised article.

Sincerely yours,

Constantin Volovat

Reviewer 3 Report

In this review manuscript, the authors have discussed the role of nanovectors for the improvement of cancer immunotherapy. The topic of the manuscript is novel and interesting. However, in my opinion, the manuscript should undergo major revisions in order to be appropriate for publishing.

  • The manuscript should be revised in terms of English language and grammar throughout. Parts of the paper is well written, whereas others contain obvious grammatical mistakes that ought to be revised.

  • Table 1; The word “Mycell” that has been used on multiple occasions should be changed to “Micelle”.

  • Table 1: Micelles mainly belong to organic nanoparticles, whereas in this Table, they have been categorized as inorganic…

  • Line 223: The authors have mentioned that the structure of the lipid nanoparticles has many similarities to the cell membrane. The authors must heed that lipid nanoparticles comprise many subcategories including the solid-lipid nanoparticles, nanoemulsions, lipid nanocapsules and liposomes. Among these, only liposomes present similarities to the cell membrane. Hence, the generalization of this statement to all lipid nanoparticles is essentially incorrect.

  • Line 298: EPR effect: The authors have explained the significance of the EPR effect for tumor targeting, but have not addressed why this is interesting from am immunotherapeutic point of view. The authors need to highlight the abundance of the suppressed immune cells in the tumor microenvironment, and the possibility of targeting those by virtue of the EPR effect. Furthermore, the concept of EPR effect is not merely applicable to the conventional nanoparticles, but also to biomimetic nanoparticles. Accordingly, the authors are encouraged to move this section to section 4; Nanomedicine to enhance immunotherapy.

  • Section 4 “Nanomedicine to enhance immunotherapy”: In my opinion, this section has to be renamed. The authors have not focused much on the potential of nanoparticles for the improvement of cancer immunotherapy. There is no discussion of the principles of nanoparticle design for passive and active targeting purposes. With one exception, there is no debate about potential targeting moieties, but rather the role of the targets in cancer development. The authors have merely described the potential targets, from an immunotherapeutic perspective, and not on the design of nanovectors themselves. Hence, this section is better called “potential targets for nanomedicine-based cancer immunotherapy”, or something similar. The subtitles should be also renamed. For instance, “tumor microenvironment as a target for NP-based immunotherapy” instead of “targeting the tumor microenvironment”, “DCs as targets for NP-based immunotherapy” rather than “targeting the DCs”, etc.

  • Section 4.2. The authors have claimed that the acidic pH and the hypoxic nature of the tumor microenvironment can be targeted by means of nanoparticles that release the cargo under the effect of these conditions. These nanoparticles, however, do not specifically target the acidic or hypoxic tumor microenvironment, but rather serve the purpose of the stimuli responsive release of the drug.

  • Line 544: “Another strategy of DCs targeting by nanomedicines is represented by RNA modulation”. RNA modulation is not a strategy of DC targeting, but rather an application of the DC targeting.

  • Section 5: The authors have failed to elucidate how this complex mathematical model for cellular communication mechanism can be applied to NP-based cancer immunotherapy, which is the main subject of the current paper.

Author Response

To : Reviewer3

Thank you very much for your suggestions. Our answers are the following (in yellow):

Comments and Suggestions for Authors

In this review manuscript, the authors have discussed the role of nanovectors for the improvement of cancer immunotherapy. The topic of the manuscript is novel and interesting. However, in my opinion, the manuscript should undergo major revisions in order to be appropriate for publishing. 

  • The manuscript should be revised in terms of English language and grammar throughout. Parts of the paper is well written, whereas others contain obvious grammatical mistakes that ought to be revised.

 We revised the English language and the grammar

  • Table 1; The word “Mycell” that has been used on multiple occasions should be changed to “Micelle”.

 We have corrected. It was a transcription error

  • Table 1: Micelles mainly belong to organic nanoparticles, whereas in this Table, they have been categorized as inorganic…

In the table, micelles belong to organic NPs. We have modified

  • Line 223: The authors have mentioned that the structure of the lipid nanoparticles has many similarities to the cell membrane. The authors must heed that lipid nanoparticles comprise many subcategories including the solid-lipid nanoparticles, nanoemulsions, lipid nanocapsules and liposomes. Among these, only liposomes present similarities to the cell membrane. Hence, the generalization of this statement to all lipid nanoparticles is essentially incorrect.

We have corrected. Thank you

  • Line 298: EPR effect: The authors have explained the significance of the EPR effect for tumor targeting, but have not addressed why this is interesting from am immunotherapeutic point of view. The authors need to highlight the abundance of the suppressed immune cells in the tumor microenvironment, and the possibility of targeting those by virtue of the EPR effect. Furthermore, the concept of EPR effect is not merely applicable to the conventional nanoparticles, but also to biomimetic nanoparticles. Accordingly, the authors are encouraged to move this section to section 4; Nanomedicine to enhance immunotherapy.

We have changed and moved in section 2. Bioactive Nanoparticles designed to modulate cancer immunotherapy

  • Section 4 “Nanomedicine to enhance immunotherapy”: In my opinion, this section has to be renamed. The authors have not focused much on the potential of nanoparticles for the improvement of cancer immunotherapy. There is no discussion of the principles of nanoparticle design for passive and active targeting purposes. With one exception, there is no debate about potential targeting moieties, but rather the role of the targets in cancer development. The authors have merely described the potential targets, from an immunotherapeutic perspective, and not on the design of nanovectors themselves. Hence, this section is better called “potential targets for nanomedicine-based cancer immunotherapy”, or something similar. The subtitles should be also renamed. For instance, “tumor microenvironment as a target for NP-based immunotherapy” instead of “targeting the tumor microenvironment”, “DCs as targets for NP-based immunotherapy” rather than “targeting the DCs”, etc.

Relating  the targeting, we have added a new part – Shortcomings in delivery and efficacy of nanoparticles

Relating the names of section and subtitles, You are right, We have changed the titles

  • Section 4.2. The authors have claimed that the acidic pH and the hypoxic nature of the tumor microenvironment can be targeted by means of nanoparticles that release the cargo under the effect of these conditions. These nanoparticles, however, do not specifically target the acidic or hypoxic tumor microenvironment, but rather serve the purpose of the stimuli responsive release of the drug.

We have corrected

  • Line 544: “Another strategy of DCs targeting by nanomedicines is represented by RNA modulation”. RNA modulation is not a strategy of DC targeting, but rather an application of the DC targeting.

 We have changed. Thank you

  • Section 5: The authors have failed to elucidate how this complex mathematical model for cellular communication mechanism can be applied to NP-based cancer immunotherapy, which is the main subject of the current paper.

We have added more explanations about this model. If is not enough, we can send you more details.

We made major revision to the article. Also, we have corrected the English spelling errors (English-editing-certificate attached).

We have attached the revised article.

Sincerely yours,

Constantin Volovat

Reviewer 4 Report

This is a potentially useful manuscript to researchers and students. However, it is riddled with Poor English, grammatical and spelling errors throughout. In addition, the authors place more than one full stop in places as well, with slight font colour changes in the text (lines 177-180). The manuscript seems rushed with little or no proofreading by the authors. While the information provided is comprehensive and the research conducted the article could benefit from a figure/scheme or two.

Specific comments:

  1. Major English revision required. Too many spelling errors and incorrect punctuations.
  2. Some information on nanovectors should be included in the introduction. The introduction could be summarized to include the major points with greater detail into each process discussed later in a separate section, for example, Cancer immunoediting (Line 75) could be briefly described in the introduction with the greater detail discussed later...
  3. In the introduction lines 51/52, it is stated "products of overexpressed tumor suppressor genes" in relation to the malignant cell. This is not correct as a malignant cell will poorly express tumor suppressors. Kindly revise.
  4. Table 1. If a single nanovector family has been used for various delivery applications, the vector could be written once under the nanovector family column, for example, PLGA. It would make the table appear neater.
  5. Under 3.1- The authors should include images of the various NPs.
  6. The authors could rephrase sentences that begin with “ Was reported”. This has been done throughout the paper. Maybe the names of the authors that have reported the information can be included in the text. This is very poor English.
  7. The authors should expand on the properties of AuNPs that make them suitable for photothermal ablation and why this would be a promising concept or an example can be included...
  8. Page 10. This reference number [284] doesn't follow the others in the section.
  9. The section on – “EPR effect as …..” does not seem to fit within the section 3 discussion on the various NPs, could follow as section 4...
  10. Page 12- Are exosomes the only EV subtype that has been used for immunotherapy? Explain and add to the manuscript.
  11. Page 14- Authors should explain what type of inorganic NPs were used?
  12. Authors should include a scheme or figure in the manuscript to avoid monotony.

Author Response

To : Reviewer 4

Thank you very much for your suggestions. Our answers are the following (in yellow):

Comments and Suggestions for Authors

This is a potentially useful manuscript to researchers and students. However, it is riddled with Poor English, grammatical and spelling errors throughout. In addition, the authors place more than one full stop in places as well, with slight font colour changes in the text (lines 177-180). The manuscript seems rushed with little or no proofreading by the authors. While the information provided is comprehensive and the research conducted the article could benefit from a figure/scheme or two.

Specific comments:

  1. Major English revision required. Too many spelling errors and incorrect punctuations.

We have revised

  1. Some information on nanovectors should be included in the introduction. The introduction could be summarized to include the major points with greater detail into each process discussed later in a separate section, for example, Cancer immunoediting (Line 75) could be briefly described in the introduction with the greater detail discussed later...

We summarized the introduction

  1. In the introduction lines 51/52, it is stated "products of overexpressed tumor suppressor genes" in relation to the malignant cell. This is not correct as a malignant cell will poorly express tumor suppressors. Kindly revise.

We have revised

  1. Table 1. If a single nanovector family has been used for various delivery applications, the vector could be written once under the nanovector family column, for example, PLGA. It would make the table appear neater.

We have changed

  1. Under 3.1- The authors should include images of the various NPs.

We have added

  1. The authors could rephrase sentences that begin with “ Was reported”. This has been done throughout the paper. Maybe the names of the authors that have reported the information can be included in the text. This is very poor English.

We have changed

  1. The authors should expand on the properties of AuNPs that make them suitable for photothermal ablation and why this would be a promising concept or an example can be included...

We have added more explanations about AuNPs

  1. Page 10. This reference number [284] doesn't follow the others in the section.

We have changed. It was a transcription error

  1. The section on – “EPR effect as …..” does not seem to fit within the section 3 discussion on the various NPs, could follow as section 4...

We have moved the section

  1. Page 12- Are exosomes the only EV subtype that has been used for immunotherapy? Explain and add to the manuscript.

We have added more details.

  1. Page 14- Authors should explain what type of inorganic NPs were used?

We have explained

  1. Authors should include a scheme or figure in the manuscript to avoid monotony.

We have added figures

We made major revision to the article. Also, we have corrected the English spelling errors (English-editing-certificate attached).

We have attached the revised article.

Sincerely yours,

Constantin Volovat

Round 2

Reviewer 2 Report

No comments.

Reviewer 3 Report

I recommend the manuscript for publication in the current form.

Reviewer 4 Report

The authors have adequately answered my comments and I have no further comments.